# Botanical, Phytochemical, Anti-Microbial and Pharmaceutical Characteristics of Hawthorn (*Crataegus*
*monogyna* Jacq.), Rosaceae

**DOI:** 10.3390/molecules26237266

**Published:** 2021-11-30

**Authors:** Federico Martinelli, Anna Perrone, Sanaz Yousefi, Alessio Papini, Stefano Castiglione, Francesco Guarino, Angela Cicatelli, Mitra Aelaei, Neda Arad, Mansour Gholami, Seyed Alireza Salami

**Affiliations:** 1Department of Biology, University of Florence, 50019 Sesto Fiorentino, Italy; federico.martinelli@unifi.it (F.M.); alessio.papini@unifi.it (A.P.); 2Department of Horticultural Science, Bu-Ali Sina University, Hamedan 65178-38695, Iran; s.yousefi@agr.basu.ac.ir (S.Y.); mgholami@basu.ac.ir (M.G.); 3Dipartimento di Chimica e Biologia, University of Salerno, 84084 Fisciano, Italy; scastiglione@unisa.it (S.C.); fguarino@unisa.it (F.G.); acicatelli@unisa.it (A.C.); 4Department of Horticulture, Faculty of Agriculture, University of Zanjan, Zanjan 45371-38791, Iran; maelaei@znu.ac.ir; 5School of Plant Sciences, The University of Arizona, Tucson, AZ 85721, USA; nedaarad@email.arizona.edu; 6Department of Horticultural Sciences, Faculty of Agricultural Science and Engineering, College of Agriculture and Natural Resources, University of Tehran, Karaj 31587-77871, Iran

**Keywords:** Hawthorn, flavonoids, pharmaceutical, phytochemical, free radical scavenging

## Abstract

Hawthorn (*Crataegus monogyna* Jacq.) is a wild edible fruit tree of the genus *Crataegus*, one of the most interesting genera of the Rosaceae family. This review is the first to consider, all together, the pharmaceutical, phytochemical, functional and therapeutic properties of *C. monogyna* based on numerous valuable secondary metabolites, including flavonoids, vitamin C, glycoside, anthocyanin, saponin, tannin and antioxidants. Previous reviews dealt with the properties of all species of the entire genera. We highlight the multi-therapeutic role that *C. monogyna* extracts could have in the treatment of different chronic and degenerative diseases, mainly focusing on flavonoids. In the first part of this comprehensive review, we describe the main botanical characteristics and summarize the studies which have been performed on the morphological and genetic characterization of the *C. monogyna* germplasm. In the second part, the key metabolites and their nutritional and pharmaceutical properties are described. This work could be an essential resource for promoting future therapeutic formulations based on this natural and potent bioactive plant extract.

## 1. Botanical Aspects

The Hawthorn species is a member of the family Rosaceae and genus *Crataegus* [1]. In the 1980s, the first studies on the intrageneric classification of the subfamily Maloideae began [2] (see Phipps et al. [3] and Campbell et al. [4]). The most recent comprehensive demographic study of the Asian and European *Crataegus* species was conducted by Christensen [5]. The genus *Crataegus* L. is one of the most important genera of the Rosaceae family and it is currently considered to comprise between 150 and 1200 species depending on the species concept employed and on the insertion of many probable taxa hybrid origins [6]. Regarding some characteristics of this genus, in the specific morphology of the leaves and seeds, the seed number and the color of the fruits, they present polymorphism, which, together with the hybridization, explain why the *Crataegus* species have different synonyms. Many species are polyploid. For these reasons, it is a highly complex species in terms of systematic botany. The identity of the common hawthorn is currently mainly referred to as *C. monogyna* Jacq., even if the taxonomy of the genus is very complex and, in the past, the species name *C. oxyacantha* L. (and *C. oxyacantha* Jacq. and other names) was employed, at least for the midland hawthorn. Currently, this last name has been rejected due to ambiguity, since it has been often used for “hawthorn” in general and hence for many species, including *C. monogyna*, the rarer *C. laevigata* (Poiret) DC. and other species with a more restricted distribution. The result of such a complex taxonomic treatment is that, in many cases, investigations on the chemical content of “hawthorn” were attributed to a species without the certainty that it was the right one.

*Crataegus pinnatifida* and *Crataegus scabrifolia* were found and used in traditional medicine in China [7]. All these species belong to the present genus polymorphism and have high levels of hybridization. This has enhanced the high number of intrageneric names and synonyms. The genus *Crataegus* is widely present in North Europe, temperate Asia, Africa and North America [5,6]. Due to the high interspecific hybridization, polyploidy and diversity [8], the taxonomy of *Crataegus* is complicated. The genus comprises from 150–500 up to 1200 species [9,10]. Climatic, geological and biological factors are associated with the characteristics of polyploidy such as latitude, altitude, habitat variety, life cycle, reproduction system, hybridization, cell size, chromosome size, chromosomal structure, sex chromosome mechanism and genotype [11].

Hawthorn is a semi-evergreen shrub or small tree, that usually has thorns, mostly growing up to 5–15 m [12,13]. Leaf morphology, seed traits, the number of seeds and the color of the fruits are highly diverse. The typical bark is smooth grey at the juvenile stage, with shallow longitudinal fissures and narrow ridges in the older phenological stage. The fruit, known as a “haw,” is similar to a berry. It is, anatomically, a pome with between one and five pyrenes having “stones” [5,14] somewhat similar to plums. The dark purplish-red pomes have a greenish calyx. Each flower has from 5 to 25 anthers, 5 sepals and 5 petals. The petals are white and pink and, usually, they are longer than the sepals. The Hawthorn fruit can be yellowish, reddish, or blackish-purple and it is usually fleshy. Each fruit has between one and five hard seeds [12]. The thorns consist of sharp-tipped branches coming from other branches or the trunk, with a length of 1–3 cm. The leaves are multi-row coils, simple, lobed and have a straight or toothed edge. They grow spirally positioned on long shoots and they are organized in clusters on spur shoots on the branches or twigs. The leaves have lobed or serrated margins and the shape is variable. The border of the flowers is at pseudo-umbrella. Each flower has from 5 to 25 anthers, 5 calyxes and 5 petals and the upper calyx does not have flowers.

Several species of birds and flowers eat and it would be essential for nectar-feeding insects. Hawthorns represent food for larvae and some Lepidoptera. Besides, haws are important for wildlife species in winter, particularly some birds which disperse the seeds in the environment. Cutting is a way of vegetative propagation of hawthorns, especially with rootless stem pieces. Small plants may be transplanted from the wild. Stratification is needed for the seeds; germination may last 1–2 years and can be enhanced if seeds are in pyrenes maintained in dry conditions at room temperature before stratification. Grafting can be performed on seedlings of other species. Characterization of morphological parameters have been performed in many species of the genus *Crataegus*, such as plant height, branches/plant, thorn number, plant canopy and berry weight and data from five randomly selected plants (in triplicate replications) were analyzed using ANOVA [15]. Intra- and interspecific variability was observed, allowing the conventional breeding for better varieties [15,16]. Morphological characterization of the *Crataegus* genus has been performed in wild regions of Ukraine using vegetative characteristics [17].

The section Oxyacanthae (containing the common hawthorn) was divided into five species subgroups based on their morphological vegetative features that correlated with their geographical distribution [17]. Hybridization and apomictic breeding often occur in the genus *Crataegus*. Indeed, there are some hybrids in the diversity centers and propagated apomictically. These interactions have produced intermediate forms, rendering the taxonomic analysis more complex [18,19]. In addition, there is high variability in the genome dimension since many species of this genus are polyploid. Some species of the genus *Crataegus* are used as ornamental plants and some parts of these plant species are edible fruits and might prevent cardiovascular diseases, diabetes and anxiety disorders [20].

## 2. Genetic Characterization

Recently, some studies have focused on the analysis of the genetic diversity of populations of hawthorn from several regions using random amplified polymorphic DNA (RAPD) [21,22,23,24] and simple sequence repeat (SSR) markers [25] (Appendix A). Coding sequences of a few genes of *C. monogyna*, such as *trnH-psbA*, ribulose-1-5-bisphosphate, carboxylase/oxygenase large subunit, translation elongation factor 1 alpha, Internal transcribed spacer (ITS) and leafy protein, have been deposited in gene banks. On the other hand, the phylogenetically related species *Crataegus pinnatifida* Bunge (Chinese hawthorn) has more than 50,000 nucleotide sequences deposited in NCBI. The sequence of the entire genome of this species is available. The main interest of this plant species is the high polyphenolic content and it has been consumed as a food and medicine source in China. Ninety-one hawthorn genotypes have been genetically analyzed using simple sequence repeat (SSR) markers [26]. A total of 265 alleles have been found from 32 SSR primer pairs. Four populations, eight sub-populations and four major clusters were identified using phylogenetic analysis.

## 3. Phytochemical Components

The fruits contain high levels of numerous valuable secondary metabolites, including flavonoid, vitamin C, glycoside, anthocyanin, saponin, tannin and antioxidant levels [27,28] and phenolic compounds [29,30,31] (Figure 1).

This plant represents a great source of bioactive molecules, including various polyphenols, such as procyanidins, epicatechin, hyperoside, isoquercitrin, chlorogenic acid, various triterpenic acids, such as ursolic acid and oleanolic acid, and other important organic molecules [32,33] (Figure 2). The parietal polysaccharides and polyphenols of *C. monogyna* have been quantitatively evaluated and the compounds can be exploited in different therapeutic applications [34].

The investigations on *Crataegus* spp. usually concentrate on detecting and quantifying flavonoids and anthocyanins that have been proven to have pharmacological effects. Hyperoside, vitexin and additional glycosylated derivatives of these compounds are the main flavonoids identified in *Crataegus* spp. [10]. Hawthorn leaves, flowers and fruits contain sugars and sugar alcohols, phenolic acids, terpenes, essential oils, phenylpropanoids, essentially hydroxycinnamic acids, lignans and flavonoids. Glucose, sucrose, fructose and xylose are sugars that have been measured in the fruits of the genus *Crataegus*. On average, the most abundant sugar in hawthorn is fructose [10].

Regarding the measurement of acids quantified in the hawthorn fruit and plant, compounds such as malic, citric, succinic, ascorbic, tartaric, quinic, protocatechin, 3- and 4-hydroxybenzoic, salicylic and syringic acids were detected [35,36,37,38]. Citric acid is the most common of the acids measured in the fruits of hawthorn plants, followed by malic acid and quinic acid. Ascorbic acid is an essential nutrient that cannot be synthesized by humans [10] and prevents oxidative stress-related pathologies such as cancer and cardiovascular disease because it acts as an antioxidant [39]. In species of the *Crataegus* genus, hydroxycinnamic acids such as chlorogenic acid, ferulic acid, coumaric acid and sinapic acid were measured. Some of these hydroxycinnamic acids have been found to act as antioxidants [40,41]. The essential flavonoids present in the *Crataegus* species are flavonol-*O*-glycoside and flavono-C-glycoside, vitexin-2′′-*O*-rhamnoside and acetyl vitexin-2′′-*O*-rhamnoside. Flavonoids own antimicrobials and antioxidant properties and stimulate antibody production [42]. Most of the experiments focused on polyphenolic compounds of another hawthorn, anatomical sections, using different extracts, which allowed identifying and characterizing numerous other groups of natural constituents with bioactive properties. However, several recent studies have mainly focused on operational processes that are very important for fostering the different health benefits. Hawthorns are an ideal source of antioxidants [43].

A study on the 56 genotypes of *Crataegus* spp. natives from different parts of Iran reported that chlorogenic acid, hyperoside and rutin were the most abundant phenols in hawthorn flower extracts [44]. The total phenolic content (TPC), the 1,1-diphenyl,2-picrylhydrazyl (DPPH) free radical scavenging activity and metal content (Zn, Fe, Cu, Mn, Cd, Cr and Pb) have been identified in wild *C.*
*monogyna* from Serbia [45].

Good yields of polyphenols extracted from hawthorn could be obtained [13]. Specifically, various chromatographic profiles revealed that both the leaves and berries of *C.*
*monogyna* are rich in isoquercitrin, vitexin, quercetin, rutin and vitexin-2- rhamnoside. Other important flavonoids in this plant are (-)- epicatechin, procyanidins B2, B5 and C1 and oligomeric proanthocyanidins with varying degrees of polymerization. The reversed-phase HPLC (RP-HPLC) technique has been employed for flavonoid quantification in flowers, leaves and berries extracts.

The flavones apigenin, luteolin and corresponding glycosides compounds such as apigenin-7-*O*-glucoside, luteolin-3′, 7-diglucoside and rutin were identified [46]. High-performance thin-layer chromatography (HPTLC) chromatographic profiles were performed using leaves and flowers of 15 *Crataegus* L. species throughout Eurasia and North America, identifying flavonoids and triterpenes. This analysis allowed the identification of fingerprinting markers usable for the pharmacopoeial species [47]. A similar technique was helpful in the qualitative and quantitative analyses of the phenolic compounds present in *C.*
*oxyacantha* and this work allowed the identification of Catechin, (-)-epicatechin and phenolic acids such as caffeic acid [48].

The high-performance liquid chromatography with diode-array detection (DAD-HPLC) was analyzed to identify rutin, quercetin-3-glucoside, caftaric and caffeic acid. A structural elucidation of quercetin, quercetin-3-*O*-β-glucoside, epicatechin, naringenin was conducted [13]. A combination of 1D, 2D NMR, MS and UV was used. Altogether, the phytochemical analysis showed that flavonoids are highly present in flowers as vitexin-2″-*O*-rhamnoside, hyperoside and oligomeric procyanidins.

For the first time, a recent phytochemical study using the extracting solvent ethyl acetate in *C. monogyna* allowed researchers to identify quercetin (1), quercetin-3-*O*-β-glucoside (2), epicatechin (3) and naringenin (4) (reported for this species for the first time). A structural elucidation of these compounds was conducted by Benabderrahmane et al. (2021) [13]. In addition, hawthorn also contains caffeoyl-quinic acid derivatives, flavonoids, coumarin, and umbelliferone. It could be used as a mother tincture for these reasons, because of the high concentration of procyanidins, although these constituents’ stability was somewhat low. The concentrations of flavonoids and umbelliferone were higher than those in the mother tincture [49].

## 4. Antioxidant Properties of Hawthorn Extracts

A correct redox balance is essential to control the signaling pathways that govern cellular functions, including the ones known to influence cell proliferation [38,50,51,52,53]. An imbalance between oxidants and antioxidants occurs when there is an overproduction of reactive oxygen species (ROS). Variations in redox homeostasis lead to the onset of oxidative stress and, consequently, the dysfunction of various regulatory mechanisms that lead to the start and development of many pathologies, including various types of cancer, atherosclerosis, cardiovascular disease, chronic inflammation and neurological disorders [52,54,55]. In this context, dietary plant constituents play a significant and positive role in countering ROS [56,57]. In particular, the intense interaction of redox-active phytochemicals maintains the redox balance that governs the pathways which control, for example, uncontrolled cell proliferation and programmed cell death. These properties have appeared to be linked to the presence of phytochemicals generally maintaining the appropriate cell redox balance through diverse mechanisms, including antioxidant activity [27]. Studies on the genus have shown that phytochemicals such as oligomeric procyanidins, flavonoids, triterpenes, polysaccharides, catecholamines and many of these compounds have been evaluated for biological functions [20,58,59,60]. In this context, several studies have revealed that the *C. monogyna* fruit represents a great source and one of the highest in terms of phenolic content with a value up to 30.63 mg gallic acid equivalent (GAE) g^−1^ in fully ripe fruit [61]. Middleton et al. (2000) [62] have reported that the content of total phenols in the ethanolic extract of dried hawthorn (*C. monogyna*) fruit is 35.4 ± 2.48 mg of GAE g^−1^ of dried mass [62].

The extracts of fruits and leaves of *C. monogyna* have unique polyphenols content, especially flavonoids belonging to the group of derivatives of quercetin and oligomeric proanthocyanidins. Hydrosolubles gels and other formulations have been prepared for topical anti-oxidant applications [63,64]. These secondary metabolites have important bioactive properties and high levels of antioxidant activity [13]. Individual compounds identified in hawthorn extracts show antioxidant properties. In addition to the previously mentioned, chlorogenic acid, epicatechin, hyperoside, quercetin also possess strong antioxidant activity [65,66,67]. This shrub does not require significant economic expenditure and represents a great source of bioactive molecules, including various polyphenols, such as procyanidins, epicatechin, hyperoside, isoquercitrin, chlorogenic acid, various triterpenic acids, ursolic acid, oleanolic acid and other critical organic molecules [33,68].

In detail, hyperoside and isoquercitrin are the main flavonols in the hawthorn berry ethanolic extract tested [67]. In addition to having active radical scavenging [69], these compounds have a vigorous anti-inflammatory activity [70]. Flavonols extracts showed a significant level of antioxidant activity in comparison to anthocyanins and other antioxidants such as ascorbic acid and quercetin [71,72].

The high content of hydroxycinnamic acids, vitexin and flavonols was measured in the flowers. On the other hand, the fruits contained high levels of anthocyanins [73]. Positive and significant correlations were observed between the phenolic mixture and antioxidant activity. However, these data also showed no significance between extract in the antioxidant capacity. Although the phenolic content of the two parts was variable, their antioxidant capacity was not statistically different. These results confirmed that *C. monogyna* might be a natural source of bioactive metabolites [41]. Recent and less recent studies have shown that extracts of leaves, flowers and fruits have high antioxidant activity [13,74].

The interactions among polyphenols from different plants with consequences on the antioxidant potential were studied. Different antioxidant extracts provide more antioxidant activity than the single ones from individual plants, confirming the positive effect of combining them [75]. The ratio-studied plants in combination with high free radical potential were investigated to optimize the maximum benefit against free radical-driven diseases.

## 5. Pharmaceutical Properties

Hawthorn (*Crataegus* spp.) is well known for its in vitro and in vivo properties beneficial for human health [76]. The green (unripe) and red (ripe) leaves, flowers and berries of *C. monogyna* are rich in bioflavonoids. A combined blend of leaves, berries and flowers was traditionally administered as an astringent, antispasmodic, cardiotonic, diuretic, hypotensive and anti-atherosclerotic agent. Various extracts of *C. monogyna* were evaluated for screening in vitro/in vivo models [32,77,78,79,80].

The molecules isolated from hawthorn are bioactive compounds with strong antioxidant, radical scavenging activity and help counteract pathologies related to oxidative stress. A summary of pharmaceutical activities against different diseases and disorders in the different human organs is shown in Appendix A.

Several pharmacological activities and properties of *C. monogyna* include hypotensive [81], hypolipidemic and antioxidant activity, [66,82,83,84,85], angiotensin-converting enzyme (ACE) inhibition [86], cardioprotective effects [87,88], anti-anxiety and anti-depression [89], adverse chronotropic and cardiotonic effects [90], protection against myocardial infarction [91], free-radical-scavenging, anti-inflammatory, gastroprotective and antimicrobial activities [67], inhibition of thromboxane A2 [92] and immunological activity [93] (Table 1).

One beneficial effect of *C. monogyna* is linked to bioflavonoids, which contribute to relaxation and dilation of the arteries, increasing the blood flow of the heart muscle and reducing the symptoms of angina and heart muscle contractions [112]. Serum cholesterol, triglycerides and glucose levels, and the number of leukocytes and platelets have been shown to decrease dramatically with the use of methanolic extract *C. monogyna* [83]. It also displays a broad variety of cytotoxic [111], anti-inflammatory [103], gastroprotective [67] and antimicrobial behaviors [13].

The immunomodulatory effect of hawthorn extract may play a critical role in the neuroprotection observed in this middle cerebral artery occlusion (MCAO)-induced stroke model [93]. Appendix A summarizes the pharmaceutical activities against different diseases and disorders in different human organs. Furthermore, the different pharmacological activities of the shrub are detailed below.

### 5.1. Protective against Cardiac Diseases

Hawthorn is the most valuable remedy for the cardiovascular system that can be found in nature. The *C.*
*monogyna* represents one of the most important medicinal plants used in antiquity. The first health effect recorded by the plant is its cardioactive effect [113]. Hawthorn extracts showed practical activities against digoxin-induced arrhythmias in rats [114].

The various protective effects of *C. monogyna* concern cardiovascular activity, attributed to flavonoids and to oligomeric proanthocyanidins (OP). The cardiac protection exerted by hawthorn is closely linked to a significant antioxidant activity [94,95,97]. OP is concentrated in the leaves, fruits and flowers and they are responsible for the pigmentation of fruits. An exciting study has clarified that the OP present in the extract of this plant reduces the oxidative stress in the myocardium after reperfusion stress and seems to inhibit apoptosis [97]. A large spectrum of hawthorn extracts’ pharmacological cardiovascular properties has been shown by many in vivo and in vitro studies, such as protective effects against atherosclerosis and vascular diseases [94]. Hawthorn more likely acts on Na+/K+-ATPase and increases the ability of cardiomyocytes to transfer calcium [115]. Hawthorn extract induces Nitric oxide (NO)-mediated endothelium-dependent vasorelaxation by eNOS phosphorylation to serine 1177 [116]. In addition, via the redox-sensitive Src/PI3-kinase/Akt-dependent phosphorylation of eNOS, WS 1442 induced endothelium-dependent NO-mediated coronary artery ring relaxation [117]. WS 1442 stimulates rbcNOS and causes NO-formation in red blood cells [118]. WS 1442 is effective in defending against endothelial barrier dysfunction [119]. Another study found that WS 1442 prevented the deleterious hyperpermeability-associated increase in Ca^2+^ ions by interference with the sarcoplasmic/endoplasmic reticulum Ca^2+^ ATPase (SERCA) and the inositol 1,4,5-trisphosphate (IP3) pathways without triggering the store-operated calcium entry (SOCE) [120].

Long-term consumption of *Crataegus* has influenced aging-related endothelial dysfunction by decreasing prostanoid-mediated contractile reactions, lowering oxidative stress and improving COX-1 and COX-2 overexpression [121]. Hawthorn fruit compounds had a significant effect in decreasing the ratio between low-density lipoprotein cholesterol (LDL-C) and serum cholesterol (TC) [122]. The tincture of *Crataegus* (TCR) successfully avoided the elevation of lipids in the serum and has resulted in a substantial decrease in lipid accumulation in the liver and aorta, reverting to the hyperlipidemic condition in rats [84]. Littleton et al. (2012) [123] showed that hawthorn leaves and flowers affect cardiac output and intravascular cholesterol levels in Zebrafish larvae used as a model for testing plant-based dietary hypercholesterolemia.

*Crataegus*, particularly the hyperoside portion, was responsible for preventing hypertension produced by L-NAME in rats and advantageous effects on the cardiovascular system [124]. The alcoholic extract of *C. monogyna* (AEC) prevents mitochondrial lipid peroxidative damage and allows the mitochondrial antioxidant state of myocardial cells to be well maintained. These studies were conducted on AEC-induced rat hearts [90]. *Crataegus* fruit extracts reduce the mitochondrial membrane potential by 1.2–4.4 mV [96]. Loss of function of cardiac mitochondrial activity is the basis for the onset of most cardiac diseases. Improving mitochondrial activity or preserving it allows the onset of pathologies related to the heart and cardiac activity to be avoided. In this context, hawthorn, thanks to the pool of bioactive molecules, especially of polyphenolic nature, allows the myocardial mitochondrial activity and consequently its functionality to be maintained. The efficacy of hawthorn flower and leaf extracts in diagnosing NYHA (New York Heart Association) Class II and III heart failure has been shown [125]. Positive results were obtained regarding the treatment of chronic heart failure together with conventional treatments.

Quantified *Crataegus* extract (CE) stimulates cardiomyogenesis from murine and human embryonic stem cells (ESC). The differentiation of cardiovascular progenitor cells induced by the pool of bioactive phytochemicals contained in CE stimulated angiogenesis and provided evidence to promote cardiovascular stem/progenitor cell differentiation. Subsequent and future experiments might be helpful, in the future, for therapeutic cardiac regeneration after myocardial infarction [126]. However, few studies have been conducted in a purely clinical setting. Other studies have confirmed the inotropic, vasodilatory and antioxidant effects of *Crataegus* extracts [127,128].

This shrub does not require high economic costs for growing. This plant extract improves muscle tone, dilates peripheral blood vessels and coronary vessels and improves the blood supply to the heart; therefore, it is considered an excellent help in treating heart disease and efficient during symptoms in the initial phase of heart failure. This clinical study demonstrated the positive effects of hawthorn extract on increased exercise tolerance in patients with class II congestive heart failure [88,129]. These clinical studies have demonstrated mainly myocardial dysfunction’s cardioprotective and preventive effects [99,109].

### 5.2. Protection against Atherosclerosis and Cholesterolemia

The first scientific studies concerning hawthorn highlighted the anti-atheromatous and coronary dilator effects [130]. CE causes a significant increase in LDL receptor activity, an increase in the excretion of bile acids and a decrease in hepatic cholesterol synthesis in rats fed an atherogenic diet. The beneficial and anti-atherogenic effects are due to flavonoids, triterpenes and cardioactive saponins and amines present in hawthorn. The mix of these bioactive molecules acts synergistically to produce the observed effects [131,132].

Studies on intestinal models using Caco-2 cells have shown that treatment with hawthorn fruit could decrease serum cholesterol linked to intestinal acyl CoA-cholesterol acyltransferase (ACAT) activity [131]. Other studies conducted on animals fed on hawthorn showed significant reductions in plasma concentrations of non-HDL cholesterol (VLDL + LDL) [100].

The phenolic compounds in CE determine the anti-atherosclerotic properties and anti-thrombotic activities [133]. The underlying mechanisms are associated with a reduced content of serum lipids, which consequently inhibit the absorption of lipids in the intestine and the de novo synthesis of cholesterol in the liver. Therefore, the reduced lipid retention causes a reduction in the number of foam cells synthesis, mainly responsible for an increase in ROS and inflammatory cytokines. CE helps endothelial integrity, including its permeability, which stops circulating lipids and macrophages/monocytes [134]. TCR successfully avoided the increase in serum and heart lipids and resulted in a substantial decrease in lipid accumulation in the liver and aorta, significantly lowering serum and cardiac lipids in rats with hypercholesterolemia [84]. TCR raised the level of antioxidant enzymes attributed to the flavonoids present in the extract. Another study showed that aerobic activity and *C.*
*monogyna* extract administration decreased ICAM-1 and E-selectin in the serum levels of 80 patients with stable angina pectoris and combination therapy was proposed as a critical complementary approach to minimize the risk of atherosclerosis and cardiac complications [135]. Polyphenolic–polysaccharide conjugates of fruit and flower extracts showed anticoagulant activity through non-direct inhibition of factor Xa, driven by antithrombin [136]. In addition, hawthorn extracts have been shown to mitigate copper toxicity towards blood cells and reduce LDL levels in a rat experiment [137].

### 5.3. Anti-Microbial Activity

*C. monogyna* showed high antimicrobial activity [74]. Both CEs (fruits and leaves) displayed significant antibacterial activity against Gram-positive bacteria and no effect against Gram-negative strains, suggesting that Gram-positive strains were more vulnerable than Gram-negative strains. Different classes of *C. monogyna* phenolic compounds, primarily flavonoids, proanthocyanidins and phenolic acids have been proposed to be the most likely active substances in inhibiting the growth of the strains of *S. aureus* [13].

Another investigation on CE showed antimicrobial activity against many microorganisms, but not against *Bacillus subtilis* and *Staphylococcus aureus* (contrarily to the previously cited study). Furthermore, it did not demonstrate any activity against the fungus *Aspergillus niger* [61]. The bactericidal action is due to natural flavonoids contained in the fruit, leaves, flowers and berry [138].

The antimicrobial potential of the berry ethanol extract of *C. monogyna* was determined by testing against Gram-positive bacteria, i.e., *Staphylococcus aureus*, *Streptococcus epidermidis*, *Bacillus subtilis*, *Micrococcus luteus*, *Micrococcus flavus*, *Lysteria monocytogenes*, *Enterococcus faecalis* and *Sarcina lutea*. In addition, Gram-negative bacteria such as *Escherichia coli*, *Pseudomonas aeruginosa*, *P. talaasii*, *Salmonella typhimurium*, *S. enteritidis* and *Proteus mirabilis*, along with two fungus strains, i.e., *Saccharomyces cerevisiae* and *Candida albicans* were shown to be sensitive to *C. monogyna* extracts using disk diffusion methods [67].

The highest antimicrobial effect of the extract was observed against *B. subtilis*, causing an inhibition zone of 23 mm, compared with streptomycin [61], which caused an inhibition zone of 37 mm. On the other hand, the plant extract did not have any antifungal effect against *S. cerevisiae* and only a moderate level of effect against *C. albicans*, while nystatin showed 20 mm of inhibition zone [61]. The berry ethanol extract of *C. monogyna* displayed significant antimicrobial activity against *Salmonella abony*, *E. coli* and *P. aeruginosa* compared to the reference (tetracycline) with inhibition zones for fully ripe fruits above 19 mm [61]. The antifungal effect of the extract that caused approximately 20 mm of inhibition zone against *C. albicans* was also remarkable.

Moreover, the leaf and berry extracts were demonstrated to have intense antimicrobial activity against *P. aeruginosa* (20 mm and 25 mm of inhibition zones, respectively) [74]. Another investigation on hawthorn extracts showed moderate bactericidal action against Gram-positive bacteria such as *Micrococcus flavus*, *Bacillus subtilis* and *Lysteria monocytogenes* and no effect on *Candida albicans* [67]. A recent study analyzed the polyphenols contained in the berry and leaves of the shrub. In addition, they also tested the antimicrobial activity of these extracts, which showed strong antibacterial activity dependently on the solvent used. The bactericidal power against Gram positive has been demonstrated in fruit extracts [13].

### 5.4. Hepatoprotective Effects

The antioxidant and hepatoprotective effects of the n-butanol extract of CE from leaves were analyzed to evaluate the protection from acute liver injury caused by Doxorubicin (DOX) administered orally to female rats. The results demonstrated that CE significantly reduced oxidative stress markers and liver enzyme concentration [102]. These results confirmed the traditional use of *C. monogyna* indicated for treating gastrointestinal disorders and protecting liver diseases.

Non-alcoholic fatty liver disease (NAFLD) is commonly present worldwide and is characterized by the high presence of fat in the liver, even in treatments. CE represents a helpful tool to counter and alleviate these disorders. The protective and therapeutic action of *C.*
*monogyna* was studied in rats fed with foods rich in fat. Although dietary enhancements from a high- to a low-fat diet improved liver damage, *C.*
*monogyna* consumption even more markedly reduced liver biomarkers and lipid peroxidation and raised antioxidant levels in rats. These data demonstrated a practical therapeutic and protective effect of *C.*
*monogyna* for non-alcoholic fatty liver disease and paved the way for future clinical studies [139]. A recent research group investigated the effect of hawthorn on carbon tetrachloride (CCl_4_)-induced liver fibrosis in rats. Lowered hepatic indices and serum enzyme markers using CE administration were observed. Furthermore, CE mitigated liver damage and fibrosis, as demonstrated by the histopathological analysis [103]. The effects were due to polyphenols such as chlorogenic acid, rutin, epicatechin and vitexin- and iso-quercetin present in herbal preparation used for rat administration.

### 5.5. Protection against Neurological Disorders

In addition, *C. monogyna* owns a high antioxidant action with inhibitory effects on Acetylcholinesterase (AchE), a property typical of medicines used to manage Alzheimer’s disease. Researchers isolated nine compounds from *C. monogyna*: β-sitosterol-3-*O*-β-D-glucopyranoside, lupeol, β-sitosterol, betulin, betulinic acid, oleanolic acid and chrysin. These compounds significantly repressed AChE. Β-sitosterol was found to be the most effective molecule as an AChE inhibitor. All these metabolites revealed potent inhibition also on butyrylcholinesterase (BChE). The mechanism of this action was investigated using docking procedures and computational prediction of the blood–brain barrier [104]. Another study evaluated the protective role of the aqueous extract of *C. monogyna* (AEOC) during acute exposure of rats to various pesticides. The combination of vitamins C and E (Vit CE) was used as the standard antioxidant. The rats showed neurological disturbances. The results showed that AECO or Vit CE significantly alleviated neurobehavioral alterations, reduced lipid peroxidation in the brain and restored endogenous antioxidant enzymes (SOD, CAT, GPx and GSH) to normal levels. Furthermore, brain DNA fragmentation and histopathology in pesticide-treated rats showed improvements after AECO administration. All results revealed that *C. monogyna* extract, rich in polyphenolic compounds, had potential antioxidant effects on rats affected by pesticide-induced oxidative brain damage [105].

### 5.6. Anti-Anxiety

Hawthorn has a very mild sedative effect on anxiety symptoms. *C. monogyna* extracts were evaluated as neurotonic components against mild-to-moderate anxiety disorders in a clinical study. The experimental work included 264 patients having generalized anxiety (DSM-III-R) with mild-to-moderate intensity. Interestingly, plant extract use was associated with anxiety improvement. These data were the basis of the commercial development of new pharmaceutical treatments against the effects of anxiety [89]. However, more research is needed before any recommendations to use hawthorn supplements as anti-anxiety can be made.

### 5.7. Against Dermatitis

The increase in the resistance against antibiotics requires the development of novel drugs against methicillin-resistant *Staphylococcus aureus*. Interestingly, *C. monogyna* extract showed to be effective in mitigating the effects of skin infections. Plant extracts showed antistaphylococcal activity associated with cutaneous disorders. The analysis was conducted using a well-diffusion assay and minimal inhibitory concentrations. These results confirmed that active extracts of hawthorn may have protective properties as antistaphylococcal metabolites, although bioactive molecules should still be identified [106].

### 5.8. Anti-Cancer

Hawthorn’s bioactive composition was tested for cytotoxic and antioxidative/pro-oxidative effects against the human laryngeal carcinoma cell line (HEp2). The results revealed significant inhibition of cell viability and radical species linked with different doses and times of use [107]. In vitro and in vivo studies showed that hawthorn could be considered a promising antitumor agent against melanoma. It can also be considered an inhibitor of tyrosinase-mediated melanogenesis and a new candidate of natural skin depigmenting agents in skincare products [51]. Homogeneous polysaccharide (HPS) extracted from hawthorn exerted anti-cancer effects on colon cancer cells through the down-regulation of the AKT pathway by decreasing phosphor-AKT and the induction of the MAPK pathway, which makes it a potential candidate in functional foods for cancer patients [140].

### 5.9. Gastroprotective Effects

CE showed anti-inflammatory properties in a rat paw edema model study. Oral administration of the extract demonstrated a significant and improved anti-inflammatory action of indomethacin, also reducing edema [67]. Gastroprotective effects were demonstrated in an acute ethanol-induced stress ulcer model in rats. CE showed significant gastroprotective activity similar to the reference drug. Furthermore, bactericidal effects on Gram-positive bacteria, such as *Lysteria monocytogenes*, *Micrococcus flavus*, *Bacillus subtilis*, have been demonstrated.

### 5.10. Anti-Diabetes

Extract of hawthorn leaves was tested for its hypoglycaemic effects when administrated to normal and streptozotocin-diabetic rats. The results revealed that hawthorn has potent anti-hyperglycemic activity modifying basal plasma insulin concentrations [108]. Ranjbar et al. (2018) [109] demonstrated significantly reduced effects of oxidative stress markers in diabetic rats at high ischemic risk after *C. monogyna* administration. A recent study evaluated the effects of CE supplementation on diabetes-induced T memory deficits and serum biochemical parameters in male rats. The results demonstrated that CE has lipid-lowering and hypoglycemic effects and improves memory in streptozotocin-induced diabetes [101]. A recent study in rats has confirmed the protective function of *C. monogyna* extracts against diabetes for treating hyperglycemia, oxidative stress and pancreatic tissue damage [141].

### 5.11. Protective against Renal Diseases

Hawthorn berries showed beneficial effects on physiological renal functions such as the excretion of urine and electrolytes. Variations in the prostaglandins E2 (PGE2) quantities were observed in a study conducted in rats. Extracts were administered and a high diuretic effect was found increasing the prostaglandins E2 and kallikrein-kinins amounts in rat blood plasma [110].

### 5.12. Genotoxic Effects

Due to scarce studies on the genotoxicity of *C. monogyna*, the genotoxic effects of fruit extract were investigated in vivo in mice. *C. monogyna* fruits extract showed weak clastogenic and/or aneugenic consequences in administrated mice, confirming that this plant extract may cause genotoxicity [142]. This evidence implies that prolonged use needs to be performed with caution. In addition, another study investigated the possible cytotoxic and clastogenic/aneugenic consequences in leukocytes and cultured liver hepatocellular carcinoma human cells and the effects on the *Salmonella typhimurium* bacterium (TA98 strains). Results showed that fruit extract of *C. monogyna* may cause genotoxic, clastogenic and aneugenic actions on cultured human cells and in vitro bacteria [111].

## 6. Conclusions

This review is the first work dealing with the pharmaceutical, functional and therapeutic properties of *C.*
*monogyna*, while previous studies have focused on the entire genus. We updated recent literature dealing with research findings dealing with these aspects. We highlighted the multi-therapeutic role that *C.*
*monogyna* extracts could have for different chronic and degenerative diseases such as atherosclerosis, cardiac diseases, cardiovascular diseases, hypercholesterolemia, hepatic disorders, neurological syndromes (i.e., Alzheimer’s disease), gastric and renal diseases, and diabetes. Extremely interesting is the complex chemical composition of diverse secondary metabolites with well-known actions against different chronic and degenerative diseases. Applying these plant extracts at different levels makes this plant extremely interesting for future therapeutic strategies based on natural metabolites. Relating to molecular mechanisms underlying the variable accumulation of these bioactive metabolites in the different tissues, sequencing projects dealing with the obtainment of the genome at least at a draft level will be necessary for the stimulation of a molecular breeding program. Now, very few genetic sequences are present in international public databases and are mainly of chloroplast origin.

Some contradictory results in several studies (particularly about the efficacy as an antimicrobial agent of hawthorn) may be related to the genus’s complex taxonomy and the specific taxonomic identity of the hawthorn itself. As a result, different responses to the same extracts may be due to different species since they are easily confounded. Therefore, clarifying the taxonomy at least of *C. monogyna* and the more strictly related taxonomic entities would be helpful to ascertain to which species the recorded several medicinal properties belong.

## Figures and Tables

**Figure 1 molecules-26-07266-f001:**
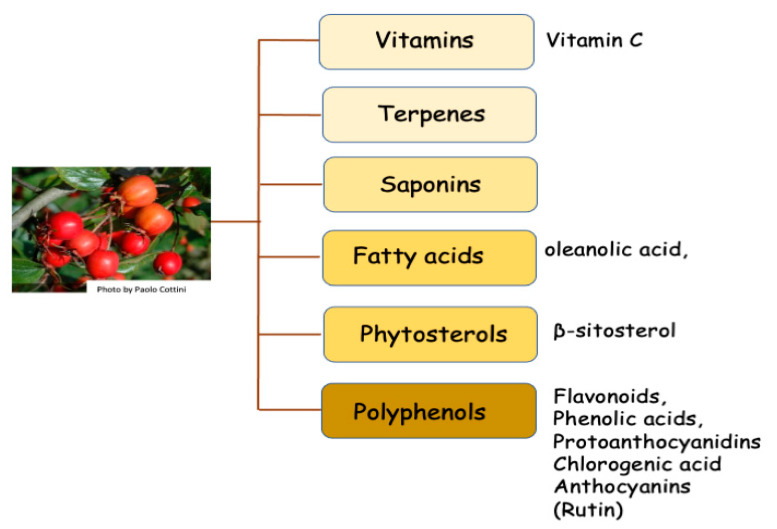
The main phytochemical components of *C. monogyna* are divided into different principal classes.

**Figure 2 molecules-26-07266-f002:**
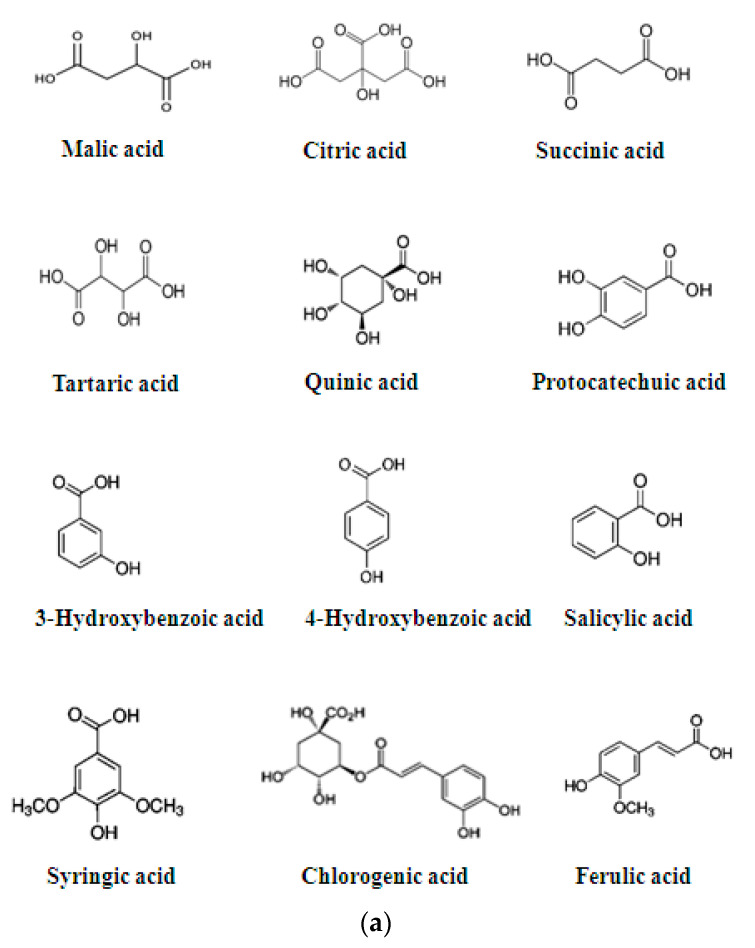
Chemical structure of acids quantified in *C. monogyna* fruits, plants and flowers. (**a**) Phenolic acids; (**b**) flavonoids and glycosylates derivates in *C. monogyna* fruits, plants and flowers; (**c**) other flavonoids: epicatechin and procyanidin; (**d**) triterpenoid compounds.

**Table 1 molecules-26-07266-t001:** Metabolic compounds, the mixture used in the study, action type, used plant tissues, type of evidence and references of research studies dealing with pharmaceutical properties of *C. monogyna*.

Metabolic Compounds or Mixture	Type of Action	Tissue	Type of Evidence (In Vitro, In Vivo, Traditional Medicine)	References
Proanthocyanidins	Cardiovascular activity	Leaves, fruits, flowers	In vitro, traditional medicine, in vivo	[94,95,96]
Proanthocyanidins	Apoptosis inhibition	Leaves, fruits, flowers	In vitro	[97]
Different plant extracts	Anti-inflammatory effects	Berries, leaves	In vivo	[96]
Different plant extracts	Myocardial mitochondrial activity	Flowers, leaves	In vitro, in vivo	[90]
Different plant extracts	Mitochondrial antioxidant activity	Flowers, leaves	In vitro	[90]
Different plant extracts	Protective against Class II and III heart failure	Flower and leaf extracts	In vitro, in vivo	[98]
Different plant extracts	Protective against hypertension, heart and digestive disorders	Berries, flowers and leaves	In vitro, in vivo	[67]
Different plant extracts	Dilatation of peripheral blood vessels and coronary vessels	Leaves and berries	In vivo	[99]
Different plant extracts	Hypo-lipidaemic, anti-inflammatory, antianxiety	Leaves and berries	In vivo	[87]
Fruits Fruits extracts	Anti-ipercholesterolemia Lipid-lowering and hypoglycemic effects (rats)	Fruits Fruits	In vivo In vivo	[100] [101]
Flavonoids, polyphenols, phenolic acids Flavonoids, polyphenols, phenolic acids (different solvents)	Anti-microbial action Anti-microbial action	Leaves and berries, Leaves and berries	In vivo In vitro In vitro	[74] [13] [13]
Fruits	Anti-microbial action	Fruits	In vivo	[61]
Leaf extracts	Hepatoprotective effects	Leaves	In vivo	[102]
Chlorogenic acid, rutin, epicatechin, vitexin- and quercetin	Protective against hepatic fibrosis	Leaves	In vivo	[103]
β-Sitosterol-3-*O*-β-D-Glucopyranoside, lupeol, β-sitosterol, betulin, betulinic acid, oleanolic acid and chrysin Different extracts	Protective against neurological disorders Protective effects again brain damage by pesticide (*wistar* rats)	Leaves Different parts of plant	In vivo In vivo Ex vivo In vitro	[104] [105]
Different extracts	Anti-anxiety	Leaves, fruits	In vivo	[89]
Different extracts	Against dermatitis	Leaves, fruits	In vivo	[106]
Different extracts	Anti-cancer	Leaves, fruits	In vitro	[107]
Different extracts	Gastroprotective effects	Leaves, fruits	In vivo	[67]
Leaf extracts Different extracts	Anti-diabetes Anti-diabetes effects (diabetic rats)	Leaves	In vivo In vivo	[108] [109]
Fruit extracts Fruit extracts	Protective against renal diseases Cytotoxic and genotoxic effects	Fruits Fruits	In vivo In vitro	[110] [111]

## Data Availability

Data are contained within the article or Appendix A.

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
