# Peer review of "Botanical, Phytochemical, Anti-Microbial and Pharmaceutical Characteristics of Hawthorn (Crataegus monogyna Jacq.), Rosaceae"

_molecules, 2021, doi:10.3390/molecules26237266_

Round 1
Reviewer 1 Report
This review is an interesting contribution. A rather exhaustive review of the literature was made regarding botanical aspect, genetic characterization pharmacological properties of hawthorn. Minor concerns have to be checked and corrected before publication.
- Abbreviations should be explained the first time they appear, even if they are obvious:
Lines 116-117 / 126: what does RAPD and SSR mean (lines 116-117)? à The authors should replace “simple sequence repeat” by SSR in line 126.
Line 186: the RP acronym should be defined
Line 190: the HPTLC acronym should be defined
Line 198: the HPLC-DAD acronym should be defined
Lines 217 / 234: ROS has to be defined at line 217 and “reactive oxygen species” removed from line 234
Line 234 / 236: GAE has to be defined at line 234 and “gallic acid equivalent” replaced by GAE in line 236.
Line 288: ACE has to be defined
Lines 296-297: OM in table 2 has to be defined
Line 359: NYHA has to be defined
Line 392: OP has already been explained at line 316
Line 397: ROS has already been explained at line 217
Line 399: TCR has already been explained at line 342
Line 476: AchE has already been explained at line 473
- The atoms implicated in covalent bonding should be written in italics, as in the example flavonol-O-glucoside. Many occurrences (lines: 166 (3 occurrences), 167, 189, 199, 202, 205, 474, and one occurrence in table 2.
- Line 62: a “g” is missing (all these species belong); Line 78: the sentence should be rewritten.
- Line 150: “organic and phenolic acids” should be clarified. Indeed, phenolic acids are organic acids.
- Line 250: the authors could give examples of solvent when writing “any organic solvent”
- Lines 334 and 335 the charge of calcium ion should be noticed as a superscript
- Figure 2 needs to be revised: several symbols (question mark) appear in panels c and d. In addition, the letters a,b,c,d in the figure panels and title need to be homogenized: they are either written in upper or lower case.
- In table 2: “iso-quercetin p”. The authors should clarify this “p” letter.
- Line 671, the authors should complete reference with volume (2) and pages (24-31).
- Keywords and general information from line 575 to 620 should be added
Author Response
Dear Editor in molecules Journal,
Thank you for the opportunity to revise our manuscript, Botanical, phytochemical, anti-microbial and pharmaceutical characteristics of hawthorn (Crataegus monogyna Jacq.), Rosaceae. Thank you very much for the useful comments and corrections provided by the reviewers. Please find below our responses to the Reviewers’ comments. We have carefully evaluated the critical comments and thoughtful suggestions, responded to them point-by-point, and revised the manuscript accordingly. Furthermore, we tried to improve the English writing. The English language was critically edited by some native colleagues.
Reviewer 1:
Comment 1. Lines 116-117 / 126: what does RAPD and SSR mean (lines 116-117)? à The authors should replace “simple sequence repeat” by SSR in line 126.
Response:
We thank the Reviewer for recommending it. We have done your comment. We have spelled both acronyms: “RAPD” (random amplified polymorphic DNA) and “SSR” (simple sequence repeat)
Comment 2. Line 186: the RP acronym should be defined
Response:
We have done your comment.
We have spelled “RP-HPLC” as Reversed Phase HPLC.
Comment 3. Line 190: the HPTLC acronym should be defined
Response:
We have done your comment. HPTLC is performance thin-layer chromatography.
Comment 4. Line 198: the HPLC-DAD acronym should be defined
Response: We have done your comment. “HPLC-DAD” is High-Performance Liquid Chromatography with Diode-Array Detection.
Comment 5. Lines 217 / 234: ROS has to be defined at line 217 and “reactive oxygen species” removed from line 234
Response:
We have done your comment.
Comment 6. Line 234 / 236: GAE has to be defined at line 234 and “gallic acid equivalent” replaced by GAE in line 236.
Response:
We thank the Reviewer for recommending it. We have done your comment.
Comment 7. Line 288: ACE has to be defined
Response:
We have done your comment. ACE is angiotensin-converting enzyme.
Comment 8. Lines 296-297: OM in table 2 has to be defined
Response: We decided to eliminate it because was not relevant.
Comment 9. Line 359: NYHA has to be defined
Response:
We have done your comment. NYHA in New York Heart Association.
Comment 10. Line 392: OP has already been explained at line 316
Response:
We thank the Reviewer for recommending it. We have done your comment.
Comment 11. Line 397: ROS has already been explained at line 217
Response:
We thank the Reviewer for recommending it. We have done your comment.
Comment 12. Line 399: TCR has already been explained at line 342
Response:
We have done your comment.
Comment 13. Line 476: AchE has already been explained at line 473
Response:
We have done your comment.
Comment 14. The atoms implicated in covalent bonding should be written in italics, as in the example flavonol-O-glucoside. Many occurrences (lines: 166 (3 occurrences), 167, 189, 199, 202, 205, 474, and one occurrence in table 2.
Response:
We have done your comment.
Comment 15. Line 62: a “g” is missing (all these species belong);
Response:
We have done your comment.
Comment 16. Line 78: the sentence should be rewritten.
Response: Done.
Comment 17. Line 150: “organic and phenolic acids” should be clarified. Indeed, phenolic acids are organic acids.
Response: “organic” eliminated.
Comment 18. Line 250: the authors could give examples of solvent when writing “any organic solvent”
Response: We eliminated it.
Comment 19. Lines 334 and 335 the charge of calcium ion should be noticed as a superscript
Response: We have done your comment.
Comment 20. Figure 2 needs to be revised: several symbols (question mark) appear in panels c and d. In addition, the letters a,b,c,d in the figure panels and title need to be homogenized: they are either written in upper or lower case.
Response: We have homogenized the figure panels writing all name in capital uppercase. We do not see any question marks in the figures.
Comment 21. In table 2: “iso-quercetin p”. The authors should clarify this “p” letter.
Response: Eliminated “p”.
Comment 22. Line 671, the authors should complete reference with volume (2) and pages (24-31).
Response: Done.
Comment 23. Keywords and general information from line 575 to 620 should be added
Response: We removed some paragraphs to reduce this general part.
Reviewer 2 Report
This review article presents very relevant information on many aspects related to the nutraceutical/functional value of hawthorn (Crataegus monogyna). However, the information included (mostly coming from references older than 10y) is not new when compared to other review articles that include most if not all aspects addressed in this manuscript (e.g. DOI: 10.31665/JFB.2018.4163 , 10.1016/S2221-1691(12)60383-9, 10.1016/j.phytochem.2012.04.006 , 10.1002/pca.1267 , 10.3390/medicina44090091 , 10.3390/nu7095361 , 10.4172/2572-0406.1000102 , 10.1111/1365-2745.13554 , 10.1111/1365-2745.13554 , 10.1016/j.aimed.2019.09.002 , 10.3389/fphar.2020.00118 , 10.1016/B978-0-12-812491-8.00041-2). With the sole intention of improving the manuscript´s uniqueness and scientific soundness, the following is suggested to the authors:
- Replace at least 30% of references older than 10 years with new evidence, including that commented on in other reviews such as those mentioned above.
- Data presentation must be improved (e.g. Table 2 should include a column of mechanisms instead of experimental model). The resolution of all figures must be improved (> 300 dpi)
- It is advised that the manuscript be sent to a formal translation agency or reviewed by a native English-spoken colleague.
- Table 1 and Figure 3 are not needed
- Discussion on health benefits must be structured in a QSAR (quantitative structure-property relationship)-type manner so that the phytochemical composition section is linked with the benefits.
Author Response
Dear Respected Reviewer,
Reviewer 2:
Comment 1. Replace at least 30% of references older than 10 years with new evidence, including that commented on in other reviews such as those mentioned above.
Response: We would like to keep the older references dealing with botanical classification of hawthorn. However, we added more than 25 references throughout the manuscript. When possible, we have eliminating some of the older ones.
Comment 2. Data presentation must be improved (e.g. Table 2 should include a column of mechanisms instead of experimental model). The resolution of all figures must be improved (> 300 dpi)
Response: Unfortunately the mechanisms of actions are mostly not well-known and still work are ongoing to clarify these mechanisms. We would like to maintain Table 2 as a sort of collection of articles dealing with the different pharmaceutical studies performed on hawthorn extract without the risk of providing wrong and unclear information of molecular mechanisms of action of these extracts. We improved the resolution of figures as much as possible.
Comment 3. It is advised that the manuscript be sent to a formal translation agency or reviewed by a native English-spoken colleague.
Response: Done.
Comment 4. Table 1 and Figure 3 are not needed
Response: Moved so supplementary.
Comment 5. Discussion on health benefits must be structured in a QSAR (quantitative structure-property relationship)-type manner so that the phytochemical composition section is linked with the benefits.
Response: Unfortunately, we are not able to perform the QSAR/QSPR structural analysis because of the unknown mechanisms of beneficial actions of many of the hawthorn metabolites experimented in the present review. This will lead to confounding and probably not correct information. Thousands of quantitative structure-activity and structure-property relationships (QSARs/QSPRs) have been published, as well as numerous papers on the correct procedures for QSAR/QSPR analysis, however, many analyses are still carried out incorrectly, or in a less than satisfactory manner. A number of 21 types of error that continue to be perpetrated in the QSAR/QSPR literature were reviewed by Dearden et al., 2009 and many other authors. Although there some recommendations for avoiding errors and for improving and enhancing QSAR/QSPR analyses, however, a comprehensive set of data and more detailed pathways are needed to eventually have a good model. We believe that there is still a lack of such data in hawthorn and current studies on model development may be associated with bias.
Round 2
Reviewer 2 Report
No further comment